# Resident Impact of the Single Site Order Restricting Staff Mobility across Long-Term Care Homes in British Columbia, Canada

**DOI:** 10.3390/healthcare11243190

**Published:** 2023-12-17

**Authors:** Farinaz Havaei, Sabina Staempfli, Andy Ma, Joanie Sims-Gould, Thea Franke, Minjeong Park

**Affiliations:** 1School of Nursing, University of British Columbia, Vancouver, BC V6T 2B5, Canada; farinaz.havaei@ubc.ca (F.H.); atf.ma@ubc.ca (A.M.); minjeong.park@alumni.ubc.ca (M.P.); 2Department of Family Practice, University of British Columbia, Vancouver, BC V6T 1Z3, Canada; simsg@mail.ubc.ca; 3Center for Hip Health and Mobility, University of British Columbia, Vancouver, BC V5Z 1M9, Canada; thea.franke@ubc.ca

**Keywords:** patient supported research, COVID-19, older adult, mental health, pandemic management, policy evaluation, implementation science, staffing, quality of care, quality of life

## Abstract

The Single Site Order (SSO)—a policy restricting staff from working at multiple long-term care (LTC) homes—was mandated by the Public Health Agency of Canada to control the spread of COVID-19 in LTC homes, where nearly 70% of COVID-19-related deaths in Canada occurred. This mixed methods study assesses the impact of the SSO on LTC residents in British Columbia. Interviews were conducted (residents (n = 6), family members (n = 9), staff (n = 18), and leadership (n = 10) from long-term care homes (n = 4)) and analyzed using thematic analysis. Administrative data were collected between April 2019 and March 2020 and between April 2020 and March 2021 and analyzed using descriptive statistics and data visualization. Qualitative and quantitative data were triangulated and demonstrated that staffing challenges became worse during the implementation of the SSO, resulting in the mental and physical health deterioration of LTC residents. Qualitative data demonstrated decreased time for personalized and proactive care, increased communication challenges, and increased loneliness and isolation. Quantitative data showed a decline in activities of daily living, increased antipsychotic medication use, pressure ulcers, behavioural symptoms, and an increase in falls. Addressing staff workload and staffing shortages during SSO-related policy implementation is essential to avoid resident health deterioration.

## 1. Introduction

The COVID-19 pandemic disproportionately affected the health of older adults [1] with particularly devastating effects on long-term care (LTC) home residents [2], where a large proportion of residents are over 65 years of age. During the first wave of the pandemic, a very high concentration of COVID-19-related mortality occurred in LTC homes across Organization for Economic Co-operation and Development (OECD) member countries [2]. Canada’s overall mortality rates for COVID-19 were lower compared to most OECD countries due in part to forceful policy measures effectively limiting community transmission [2], yet mortality rates among residents of Canadian LTC homes were shockingly high [3]. After the first six months of the pandemic, nearly 70% of COVID-19-related deaths in Canada occurred in LTC homes, which is significantly higher when compared to the international average of 41% [3].

In response to high mortality rates in LTC homes, the Public Health Agency of Canada (PHAC) mandated several pandemic management strategies to curb the spread of the virus and protect the health and safety of LTC residents and staff [4]. Pandemic management strategies included visitation restrictions, mandated use of personal protective equipment (PPE), including face masks and face shields, COVID-19 screening protocols prior to entry, and policies, including the one preventing care providers from working across multiple sites (i.e., the Single Site Order [SSO]) [4]. Although preliminary evidence indicates pandemic management strategies reduced the spread of COVID-19 [5,6], the complexity of the policies resulted in unintended consequences to the health and well-being of LTC residents, staff, and family members [5,7]. The purpose of this study (as a part of the larger impact evaluation) is to evaluate the impact of the SSO on residents of four LTC homes in one Canadian province, British Columbia (BC).

### 1.1. LTC in Canada and International Context

In Canada, LTC homes are residential care settings for older adults that offer 24 h on-site personal care, medical care, and housekeeping services [8]. As of March 2021, there were 2076 LTC homes in Canada, of which just over half (54%) are privately owned (29% of which operate as for-profit institutions) and 46% are publicly owned [8]. The LTC sector exists outside of the Canada Health Act—Canada’s federal legislation providing publicly funded universal health care coverage—and is thus not federally insured [9]. As a result, there exists great variability in funding structures and legislation across provinces and territories [10].

Similar to the demographics of other OECD country LTC homes [11], most residents in Canadian LTC homes are over 80 years old [12] and have complex healthcare needs (frailty, require assistance with activities of daily living (ADLs), multi-morbidity) [8,12], and roughly three-quarters of all residents live with moderate to severe cognitive impairment [13]. Between 75 and 85% of all direct care for Canadian LTC home residents is provided by care aides, an unregulated workforce that is largely made up of middle-aged women with a high school diploma who speak English as their additional language [14]. Prior to the onset of the pandemic, nearly a quarter (24.3%) of care aides reported working at more than one LTC home [15] due largely to the inability to earn a living wage from working as a care aide in a full-time job, often necessitating an additional part-time job due to the low pay and often precarious employment situations without benefits or sick leave [12]. As the global population ages, LTC homes must accommodate increasingly medically complex residents and increasing numbers of residents living with cognitive impairment as the population ages, which requires a skilled and consistent workforce [10]. Most OECD countries have a similarly underpaid, unregulated, insufficiently educated (for the complexity of the work) workforce providing care in LTC homes [11]. Canadian care homes increasingly struggle with attracting and retaining trained workers [10], as is the case in most other OECD countries [11]. The impetus for the PHAC to implement the SSO was ultimately due to the combination of existing staffing challenges [10], the transient nature of workers employed at multiple LTC homes [15], the medically vulnerable residents and the high transmission risk of COVID-19 in crowded LTC homes [2]. Other countries similarly implemented policies restricting staff mobility across LTC homes in attempts to curb the spread of the virus [16].

### 1.2. Single Site Order 

BC was the first province in Canada to implement the SSO [12] and has been widely acknowledged for the rapid mobilization of emergency measures implemented to curb the transmission of the virus [5]. The BC SSO (mandated on 10 April 2020) falls under the Health Care Labour Adjustment Order restricting most staff (e.g., nurses, care aides) working in LTC homes in BC to one single site [17]. Pharmacists, nurse practitioners, physicians, and dieticians were excluded from the SSO [17]. The SSO was designed to limit transmission of the virus and stabilize the front-line workforce by bringing care home staff under the employment of the BC government and providing staff with full-time pay with benefits [12]. By 18 June 2020, more than 8700 multi-site staff at 5000 LTC homes in BC were reassigned to single sites [17]. The SSO did not limit staff from working in acute care or the home healthcare sector [17]. Although the SSO is attributed to the lower infection and mortality rates in BC compared to other jurisdictions [5], its impact on LTC resident outcomes has not yet been studied. Our objectives of this study are to determine the psychological and physiological health and safety impact of the SSO policy implementation on key resident outcome indicators.

## 2. Materials and Methods

This mixed-method study was part of a larger study examining the overall impact of the SSO. This study analyzed resident outcome administrative data in conjunction with semi-structured interviews with residents, family members, staff, and leaders in four partner LTC homes in a moderate to high-density urban area in BC. The four LTC homes span three municipalities (Vancouver, Richmond, Surrey) and two health authorities (Vancouver Coastal and Fraser Health). The LTC homes are described in Table 1. A steering committee of leadership, nursing staff, support workers, family members, and resident representatives from either the four partner LTC homes or a representative health authority was formed. The committee met regularly to inform the direction of the research and consulted on each stage of the research process. Ethics approval was obtained from Research Ethics Boards of the University of British Columbia, the four partner LTC homes and their respective health authorities (H20-03967).

### 2.1. Qualitative Methods

**Recruitment and sampling.** Purposive sampling was used to recruit 35 interviewees across four partner LTC homes (Table 1). Purposive sampling allows researchers to ensure representation of specified categories of individuals with unique and important perspectives [18]. Interview participants included LTC home residents (n = 6), family members (n = 10), staff (n = 18), and leadership (n = 10). Inclusion criteria were broad, and no exclusion criteria were specified to ensure a diverse sample. Participants were eligible if they spoke English or Cantonese and had been living, working, or had family members living in one of the four partner LTC homes for any period of time during the pandemic. All participants spoke English or Cantonese and provided written and verbal consent to an audio-recorded interview. Staff supported the recruitment of residents and family members, and leaders assisted with the recruitment of staff. Leaders were contacted directly or referred by other leadership team members. All participants who consented to participate were interviewed, and none revoked their participation after interviews were conducted.

**Data collection.** Semi-structured interviews were conducted from February to April 2021 by researchers with previous academic interviewing experience (JSG, SS, TF). A semi-structured data collection technique is able to give a guiding structure to interviews while eliciting meaning from the lived experience of interviewees, which is integral to the phenomenological theoretical approach used in this research [19]. Interview guides (designed for each participant group) had a broad focus on evaluating the impact of the SSO and were vetted by the steering committee. Interviews were conducted virtually (in Zoom, a secure video conferencing platform). No repeat interviews were conducted. One interviewer (JSG, SS, TF), and one note taker (VW, VM) (all female) participated in each interview. Some of the research team members (JSG, FH, SS, TF) had previously established relationships with interview participants through previous research collaborations. Interviews were started with interviewers clearly describing the research goals. Staff or family members assisted residents in setting up video calls in a private room. Interviews were 30–60 min and were recorded, transcribed, and anonymized. Transcripts were not returned to participants or provide feedback on findings. 

**Data analysis.** Using a theoretical perspective of descriptive phenomenology, a thematic analysis of interviews was conducted following the iterative 6-step framework of Braun and Clarke [20]. To ensure trustworthiness (credibility, transferability, dependability, and confirmability) of the analysis, Nowell et al. [21] were used as a guide to ensure rigour and quality of data analysis. To familiarize themselves with data (Step 1), two team members (TF, SS) read through all interview transcripts and interview notes. After discussions at a team meeting (SS, TF, JSG), initial codes were systematically generated (Step 2). An initial thematic framework was created (based on the larger implementation science study following the Consolidated Framework for Implementation Research domains) and used to guide interview coding (TF, SS) using NVivo 12 software. Full paragraphs were coded so as not to lose contextual meaning. New codes were identified, all relevant coded data was collated, and sub-themes were identified (Step 3). Themes and sub-themes were reviewed and refined, validity of themes was considered in relation to the whole data set, and the thematic framework was refined (SS). Meetings were held (SS, TF, AM, FH) to discuss whether the revised thematic map accurately reflected meaning in the dataset (Step 4), and coding was stopped as refinements were determined not to add further substance. Each theme was analyzed, and stories with accompanying narratives were added (SS) (Step 5), which were triangulated with quantitative data and collated into a thematic framework (Figure 1) and report (Step 6).

### 2.2. Quantitative Methods

**Sample and data collection.** Aggregated resident data from four partner LTC homes were extracted from the Canadian Institute of Health Information. The first four extracted quarters (April 2019 to March 2020) were defined as ‘pre-COVID-19’. The following period of four quarters (April 2020 to March 2021) was defined as ‘during COVID-19’. The data collected were considered ‘aggregated’ because each datum contained a quarterly mean value per LTC home, summarizing information about hundreds of residents for the quarter.

Resident indicators include demographic variables (sex, age, and length of stay), resident behaviour (social engagement and aggressive behaviour), and a series of risk-adjusted quality indicators (functional dependence, worsening behavioural symptoms, antipsychotic medication use, falls, worsening pressure ulcers, physical restraint usage, and weight loss). These indicators were selected to cover a variety of resident outcomes. Most indicators were calculated based on assessed or discharged residents, while others were computed based on total active residents in an LTC home. Table 2 provides a list of resident indicators and their operational definitions.

**Data analysis.** Data analysis comprised descriptive statistics and data visualization. Due to the aggregated nature of the resident data available and resultant limited sample size (each LTC home only had one datum per time point), descriptive methods were deemed more appropriate than inferential statistics. Scatterplots were created for each indicator, to descriptively illustrate the change across eight quarters from April 2019 to March 2021. Each scatterplot was supplemented with a two-part linear trendline that illustrated the overall pre-pandemic and during-pandemic trends for that indicator. These trendlines display lines of best fit as computed by ordinary least-square methods. Trend slopes were also annotated to the plots to facilitate comparison of pre-pandemic and during-pandemic trends.

## 3. Results

### 3.1. Qualitative

Table 1 above describes interview participant characteristics. Two overarching categories emerged from interview data associated with SSO implementation: factors protecting residents from mental and physical health deterioration (blue boxes Figure 1) and factors contributing to the mental and physical health deterioration of LTC home residents (pink boxes Figure 1). 

**Factors protecting residents from mental and physical health deterioration.** SSO implementation was associated with two major themes that were perceived by participants to protect against mental and physical health deterioration of LTC home residents: (1) consistency in care and (2) a sense of safety. 

**Consistency in care**. The quality of care was perceived by families and residents to be unchanged during SSO implementation. Family and residents from all four partner LTC homes reported high satisfaction with the care they received during this time. Family, staff, and leaders noticed an increased consistency in staff members due to the decreased mobility of care staff (due to the SSO), which promoted resident recognition of staff members, staff member familiarity with particularities of resident routines, and created environments that fostered staff and resident connections. Consistency in care was especially beneficial for residents with cognitive impairment and residents with complex routines and individualized care needs. 


*I’m very much for the [SSO]. Especially for any of the residents [with] dementia. The routine is so so important and knowing who it is [giving the care]. Every so often my father [resident] will say, ‘well, I don’t know you’ to the [caregiver], ‘why are you giving me pills’, and ‘what pills are you giving me’. … one of his nurses [had] asked me if I noticed any difference in [my father] and [the nurse] says ‘well he’s becoming more resistant to taking medications’. … On his yearly Parkinson’s visit with [the doctor] … the nurse who most often does the night shift with [resident] stayed on after her shift so that she could talk with [the doctor to explain those changes]. It’s just that level of care that having that individual who knows the resident is so important, in my opinion. [Family Member 2, The Manor]*


**Sense of safety.** Family, residents, and staff felt the SSO provided residents with an increased sense of safety and security, which protected their mental health during SSO implementation. Reducing the risk of exposure to COVID-19 by limiting the mobility of staff across LTC homes reduced the fear, anxiety, and vulnerability associated with potentially contracting the virus. Residents and family members highly valued feeling protected by LTC homes, often rating their sense of safety as the top benefit of living in an LTC home. 


*During [the first wave], there were a lot of LTC homes that had fairly widespread [infection], they had to go through lockdown … and it was a very terrifying experience for families. So, [the SSO] is a very welcoming decision. … we’re dealing with the vulnerable aging population… I think the one site policy is the best policy. [Family Member 2, Rosewood]*


**Factors contributing to resident mental and physical health deterioration**. SSO implementation was associated with one overarching theme of staffing challenges that was perceived by participants to contribute to the mental and physical health deterioration of LTC home residents. Three subthemes related to staffing challenges also contributed to the health deterioration of residents, including (1) decreased time for providing personalized and proactive care, (2) communication challenges, and (3) loneliness and isolation. The implementation of other pandemic management strategies amplified staffing challenges, and all three subthemes further contributed to the health deterioration of residents. 

**Staffing challenges.** As per leadership and staff, the number of available casual employees for all four LTC homes substantially decreased during SSO implementation. Leadership describes the loss in a casual pool as particularly impactful for direct care staff. Many employees retired early or quit at the onset of the pandemic. This shortage in staff led leaders to increase the amount of overtime for existing staff (to compensate for unfilled positions). Overtime increased the pressure on existing staff (who worked more hours) and led to increasing burnout and staff absenteeism. Leaders tried to compensate for the loss in staff by hiring new staff, but time spent training new staff increased the burden on existing staff. 


*When [care staff] work a lot of overtime because of lack of staff, staff get burned out and you can’t fill the overtime slots anymore. [This] increased [the] frequency [of] calling in sick too. [Leader 2, Lake Bay]*


The mass hiring of new staff increased care challenges for residents with cognitive impairment. One family member *[Family Member 2, The Manor]* explains their father, who has Parkinson’s, started to refuse to take his medication because he did not know or trust the person giving it to him, which according to the family member, caused his condition to worsen. The residents noticed staffing challenges as well. One resident described being hesitant to press the call bell to ask for help for fear of increasing the burden on staff. 

**Decreased time for personalized and proactive care.** Staffing challenges decreased the time available for staff to provide personalized care, attention to detail, and proactive (as opposed to reactive) care for residents, which contributed to the deterioration of resident health. One family member (Family Member 1, The Manor) described how her father had recurring dehydration until they realized that care staff would bring him a juice box or water without a straw inserted. The resident subsequently could not drink but would forget to ask for caregivers to insert the straw for him (due to cognitive impairments). Another family member stated that although she considered the care to be “excellent”, lack of time for personalized care negatively impacted her father (resident) both physically and mentally:


*There [are] only so many [residents] that the clinical staff can care for in a given day. There’s only so much they can do and honestly coming back here [after not visiting due to visitation restrictions] … he had broken eyeglasses [and] no hearing aids. [His] hair wasn’t cut. So many details that aren’t life threatening but are many activities of daily living were not to the standard that he had been accustomed to. [Family Member 4, Lake Bay]*


**Communication challenges**. The leaders of one culturally specific LTC home had difficulty filling their casual staffing pool due to the language requirements of residents. Eventually, the leaders had to rely on agency staff to fill gaps in the care aide staffing pool, but agency staff did not speak the main language of residents at the LTC home. The residents had difficulty expressing their needs to care providers, which affected their mental and physical health. 


*Agency [staff] that don’t speak the [same] language, it will be very difficult to know the needs of the resident that in turn makes the resident very anxious when they are not able to express their needs or being understood about what they want. [Other staff] will be shouldering or [bearing] the responsibility of our work happening [with the] resident, so it takes some time for the agency staff to know the procedure, [to] start learning a little bit of very basic [language] in order to communicate. [Leader 2, Rosewood]*


**Increased loneliness and isolation.** Many residents expressed the devastating effects of the loss of human contact during lockdowns, brought on by the lack of time available for care due to SSO-related staffing challenges. The residents expressed the burden of loneliness and isolation as feeling punished or emotionally tormented.


*I think [the reason for decline in residents is] the lack of interaction with the world, [the] isolation. … One lady she wears pearls and in June she’s walking around she always sits in the same chair and [says hello to me, but] when [I returned to the facility I noticed] she’s just gone downhill so quickly. You notice people disappearing, … If you’re really paying attention, you know that the residents are suffering. [Family Member 1, Lake Bay].*


Staff felt an immense sense of responsibility to go above and beyond their defined roles to try to offset the negative health impacts of loneliness and isolation. However, these actions further increased staff workload. One staff member came into the LTC home on their weekend off to paint the nails of residents. Another staff member purchased Japanese television for a resident for a year because they only spoke Japanese and could not interact with any other residents.


*[When] nobody could move and gather [for performances], the recreation staff would go to their [assigned] particular floors. … Then [the recreation team] were entertaining even more than before [the pandemic] to make [residents] happy not lonely. And the LPNs also [helped with] phone calls from family members on their birthdays and [the nurses would] dress [up the residents] and bring [them] to the windows and they’ll go in the parking lot and [get] birthday wishes and balloons [the residents] can see [their family from] afar. As well, the pastor who’s helping to do one thing or another [for residents]. [Staff Member 0, Seaside]*


**Concurrent implementation of other pandemic management strategies.** Each LTC home concurrently implemented other pandemic management strategies for infection prevention and control (IPAC) alongside the SSO. Three IPAC strategies, including visitor restrictions, decreased recreational programming, and PPE use, amplified SSO-related staffing challenges and contributed to the mental and physical health deterioration of residents. 

Family members provide not only companionship to residents (**addressing loneliness and isolation**) but also act as advocates for **proactive care**, provide essential **personalized care** such as feeding, grooming, and cleaning, and provide avenues of **communication** [22]. Visitor restrictions created a gap in care for residents, which increased the burden on staff who had to step up to provide missing essential care, and increased **staffing needs** beyond the baseline (which were already exacerbated due to SSO-related staffing challenges).


*All of a sudden, my staff had to be the family in many ways and that’s the only reason we staffed up because I don’t have that person that came in every day to help them have lunch. Somebody has to do [it]…the staffing has been a real challenge. [Leader 1, Seaside]*


IPAC distancing requirements decreased the frequency and scope of recreational programming, where programs had to be run with fewer residents at a time, decreasing accessibility and increasing **staffing requirements**. Decreased programming combined with SSO-related staffing challenges amplified the health deterioration of residents who experienced more **loneliness and isolation** due to decreased physical and social contact with other residents and staff. One resident (*Resident 1, The Manor*) describes missing Bingo games, musical groups, and the companionship that came from these programs. She stated she had to rely on books to keep her occupied and conceded that every day was a long day. 

Face coverings from PPE mandates amplified SSO-related **staffing challenges** (training burden, additional workloads to assume) and increased **communication challenges** (face covering hides expressions and dampens voices), especially between staff and residents with cognitive impairment.


*It was really challenging. …when you try to talk nobody hears you so it’s so hard to have to start raising your voice and the communication is impaired. Even if I smile to the elders or to my team, nobody can see. [Staff Member 0, Seaside]*


PPE donning and doffing also impacted the speed with which staff could assist residents when alarms sounded (which ring when a resident at risk of falling is climbing out of bed).


*When we were in outbreak… all the residents had to be isolated in their own suite. That’s the time that we found increase [in] falls. And given the fact that [staff must don and doff] before we can reach [residents], even if we hear the bed alarm ring. In fact, the residents do not really have [anything else to] do other than staying [in their] room, so they get bored they get up and they didn’t have [access to] exercise so [they fall]. [Leader 1, Rosewood]*


### 3.2. Quantitative

The aggregated demographic variables of age and sex were calculated per LTC home based on the total active residents (residents with admission, assessment, or discharge records from that quarter); the mean length of stay was calculated based on each LTC home’s discharged residents of that quarter. The overall average for mean resident age, across all LTC homes and quarters, was 84.7 years (SD = 2.8). All LTC homes had a greater proportion of female residents than males, with an average female resident proportion of 61% (SD = 7.6). The overall average resident stay length was 3.2 years (SD = 1.1).

An overview of trends for resident behaviour scores and quality indicator proportions, before and during the pandemic, is shown in Figure 2 above, through trend statistics (overall averages and trend slopes). Overall trends and trends by LTC home are also visualized in scatterplots. As indicated by the trend overview, there was an increasing trend in overall the pre-pandemic Index of Social Engagement (ISE) scores. However, the rate slowed down during the pandemic. The average score across LTC homes rose by 0.15% per quarter but stabilized during the pandemic, with an overall increase of 0.06% per quarter. Quality indicator trends suggest that antipsychotic medication use worsened the most during the pandemic. The overall proportion of assessed residents that had taken antipsychotics without diagnoses of psychosis had been slightly increasing before the pandemic (+0.11% per quarter) but increased across the pandemic onset and continued increasing at a relatively much higher rate during the pandemic (+1.53% per quarter).

A similar pandemic-exacerbated rate of increase is shown in the worsened mid-loss ADL indicator trends. The overall proportion of assessed residents that worsened in mid-loss ADL increased at a rate of +0.10% per quarter before the pandemic but increased at a faster rate of +0.63% per quarter during the pandemic.

In the quality indicators of **worsened behavioural symptoms**, **falls in the last 30 days**, and **worsened stage 2–4 pressure ulcers**, the trend slopes demonstrated decreasing pre-pandemic trends followed by increasing trends during the pandemic. The overall proportion of assessed residents with **worsened behavioural symptoms** decreased at a rate of −0.17% per quarter pre-pandemic but increased at +0.24% per quarter during the pandemic. The overall proportion of assessed residents with **falls in the last 30 days** decreased at a rate of −0.35% pre-pandemic and increased at +0.45% per quarter during the pandemic. The overall proportion of assessed residents with **worsened stage 2–4 pressure ulcers** declined pre-pandemic (−0.11%) and slightly increased during the pandemic (+0.06%). However, the overall average proportion of assessed residents with worsened stage 2–4 pressure ulcers was notably higher across the pandemic quarters (4.9%) compared to pre-pandemic (3.9%) due to a rapid increase and decrease within one LTC home that was not captured by the slope of the linear trend.

Inversely, two quality indicators demonstrated increasing trends pre-pandemic and decreasing trends during the pandemic. The overall proportion of assessed residents with **daily physical restraints** increased at a rate of +0.17% per quarter pre-pandemic, rose across the pandemic onset, and decreased at a rate of −0.06% during the pandemic. The overall proportion of assessed residents with documented **weight loss** had an increasing trend before the pandemic (+0.24% per quarter) and a decreasing trend during the pandemic (−0.75% per quarter). Weight loss was the only quality indicator that had an overall average proportion across the pandemic quarters (7.9%) that was lower than the pre-pandemic average (8.7%).

While the overall trend statistics captured general exacerbating effects during the pandemic, by-home data indicated that the pandemic effects were not uniform. For example, by-home data for **antipsychotic usage** shows that the LTC home with the consistently lowest pre-pandemic usage proportions had a more severe rate of worsening on that quality indicator during the pandemic. For **worsening stage 2–4 pressure ulcer** rates, the LTC home with an improving pre-pandemic trend reported a gradual reversal of pre-pandemic improvement across four pandemic quarters, while another home with a worsening pre-pandemic trend reported a rapid peak in ulcer rates in the second pandemic quarter. As LTC homes vary in characteristics such as resident population, needs, and staffing resources, by-home indicator trajectories also highlighted that the effects of the pandemic varied between facilities and across time.

## 4. Discussion

Our data support emerging evidence that the SSO notably exacerbated existing staffing challenges [5,23], which contributed to both mental and physical health deterioration of residents. Staffing challenges decreased the time available for staff to provide attentive detailed care and decreased proactive care (which prevents adverse resident health outcomes). Other pandemic management strategies (visitation restrictions, decreased programming, increased PPE use) amplified staffing challenges by increasing staff workload by expanding programming (to accommodate distancing and isolation requirements) as well as existing roles to fill gaps due to decreased family presence. Communication challenges (due to face coverings and decreased family presence) also exacerbated staff workload and worsened resident outcomes due to the decreased ability of staff to provide proactive care. Administrative data reflected a deterioration of most resident outcomes, which are also related to staffing challenges. We hypothesize that providers relied on the increased use of antipsychotic medication to mitigate staffing challenges and residents’ behaviour symptoms exacerbated by increased loneliness and isolation. Staffing challenges could also explain an increase in falls, with fewer staff able to adequately meet resident mobility needs, and donning and doffing mandated PPE increased the staff’s response time to urgently assist residents when bed alarms or call bells went off. The association between staffing shortages and adverse resident outcomes is very well documented in the pre-pandemic and pandemic literature. For example, a 2022 systematic review of 11 research papers found that increased nursing hours per resident per day were associated with a statistically significant reduction in adverse resident outcomes such as pressure ulcers and urinary tract infections [24].

Although each isolated staffing issue (e.g., frequently working short-staffed, increased staff burnout and absenteeism, a high number of new staff) might not have directly impacted residents prior to the pandemic (due to most LTC homes’ ability to accommodate semi-predictable staffing fluctuations), the cumulative effect of staffing challenges created by the SSO manifested in resident health deterioration. When the staffing challenges (long standing and induced by the SSO) were combined with the concurrent implementation of other pandemic management strategies, we believe the effects were compounded, and residents suffered the consequences.

The link between staffing challenges, resident health deterioration, and the SSO is important evidence for international crisis policy making; staffing challenges are modifiable factors that could potentially mitigate the harm of LTC resident health deterioration and can be managed with comprehensive policy planning. Countries that implemented SSO-equivalent policies (preventing staff mobility across LTC homes) but that focused early and aggressively on addressing staffing shortages and increased workloads reported lower levels of COVID-19-related mortality in LTC homes (e.g., Denmark and Germany) [2]. 

For example, Germany adopted a multifaceted approach to addressing LTC staffing issues by enhancing IPAC training, increasing the minimum wage for LTC staff, recruiting students with financial and training incentives, lifting maximum working hours, making emergency childcare available to LTC workers, providing support to unpaid caregivers including family members, increasing financial incentives to study LTC-related professions, prioritizing care-related professions applying to work in the country, and making surge-staffing available for LTC homes in outbreak [16]. Although Germany also reported pre-pandemic staffing challenges [16] similar to Canada, the early and aggressive staffing-related measures most likely decreased negative resident outcomes. 

Many of the emergency staffing measures implemented by countries like Denmark and Germany were already in place pre-pandemic and were merely augmented when it became clear that staffing inadequacies and workload worsened at the onset of the pandemic and during the implementation of SSO-equivalent policies [16]. The Canadian LTC sector has long struggled to maintain adequate staffing at non-crisis levels, which has been exacerbated by systematic reductions in regulated care providers and a systematic failure to support the resilience of direct care staff, leaders, and managers [10]. Without existing emergency staffing measures in place [10], Canada was unable to provide an adequate structured response to staffing challenges, which likely resulted in the exacerbation of negative resident outcomes. Creating an environment where contingency measures are already in place for policymakers and LTC leaders is essential to enable quick and effective crisis decision making and ensure risks to the health of LTC residents are optimally mitigated. Countries (including Canada) implementing SSO-equivalent policies must take a proactive approach to improving staffing conditions in LTC by concurrently implementing strategic measures to improve staffing conditions alongside SSO-equivalent pandemic management policies. Specific policy intervention recommendations are beyond the scope of this research and are an important next step to addressing staffing challenges. Beyond the implementation of pandemic policies, our results also serve as a warning in the context of an aging global population whose care requirements will only increase over the next decades. Experts agree the first and most urgent action is to determine optimal staffing levels and appropriate staffing mix using empirical data related to resident needs [10] and improve working environments for LTC staff to improve staff retention [11] because it is LTC residents who will ultimately suffer the consequences if staffing challenges are not addressed.

## 5. Limitations 

The case-based nature of this study was not designed to create generalizable results. It is possible that interviewees from other LTC homes experienced the SSO differently. However, the detailed descriptions provided regarding our study settings (e.g., the physical, social, and political contexts) allow conclusions to be transferable to other settings and circumstances [21]. Although we were unable to conduct interviews and observations on-site (due to pandemic restrictions), we ensured the credibility of the results [21] with prolonged data engagement by multiple researchers, triangulating datasets, and enacting the advisory committee acting as an external check on the research process [21]. We increased the dependability of results by outlining the thematic analysis process clearly, providing decision and audit trails, and engaging a reflexive note taker during interviews. We also ensured qualitative results were effectively and comprehensively reported using the consolidated criteria for reporting qualitative research (COREQ) checklist, which can be found in Appendix A [25].

## 6. Conclusions

Subject matter experts predicted a significant staffing crisis before the SSO was implemented [15]. As such, it is unsurprising that the implementation of the SSO exacerbated existing staff challenges in the LTC sector. What was not known, however, was the extent of the impact of SSO-related staffing shortages on residents. The quality of care provided to residents suffered under the staffing challenges exacerbated by the SSO, which exemplified the mental and physical health deterioration of residents. LTC home leaders, decision makers, policymakers and advocates must push for measures to address staffing challenges and ease staff workloads to avoid the decline of resident health in LTC homes. Our results apply not only during crisis policy implementation but should serve as a warning to stimulate action to address current staffing issues to prevent any further health deterioration of LTC residents. 

## Figures and Tables

**Figure 1 healthcare-11-03190-f001:**
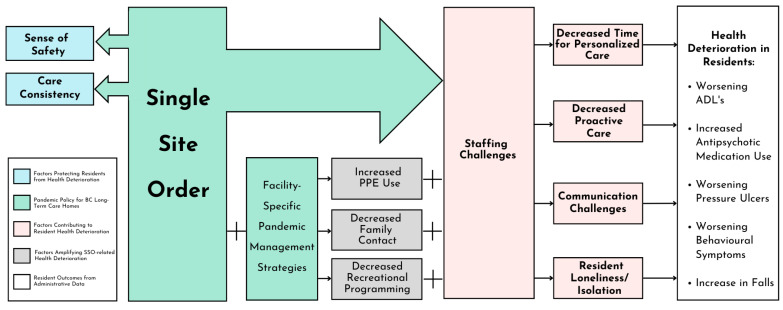
Thematic framework of qualitative indicators and quantitative results.

**Figure 2 healthcare-11-03190-f002:**
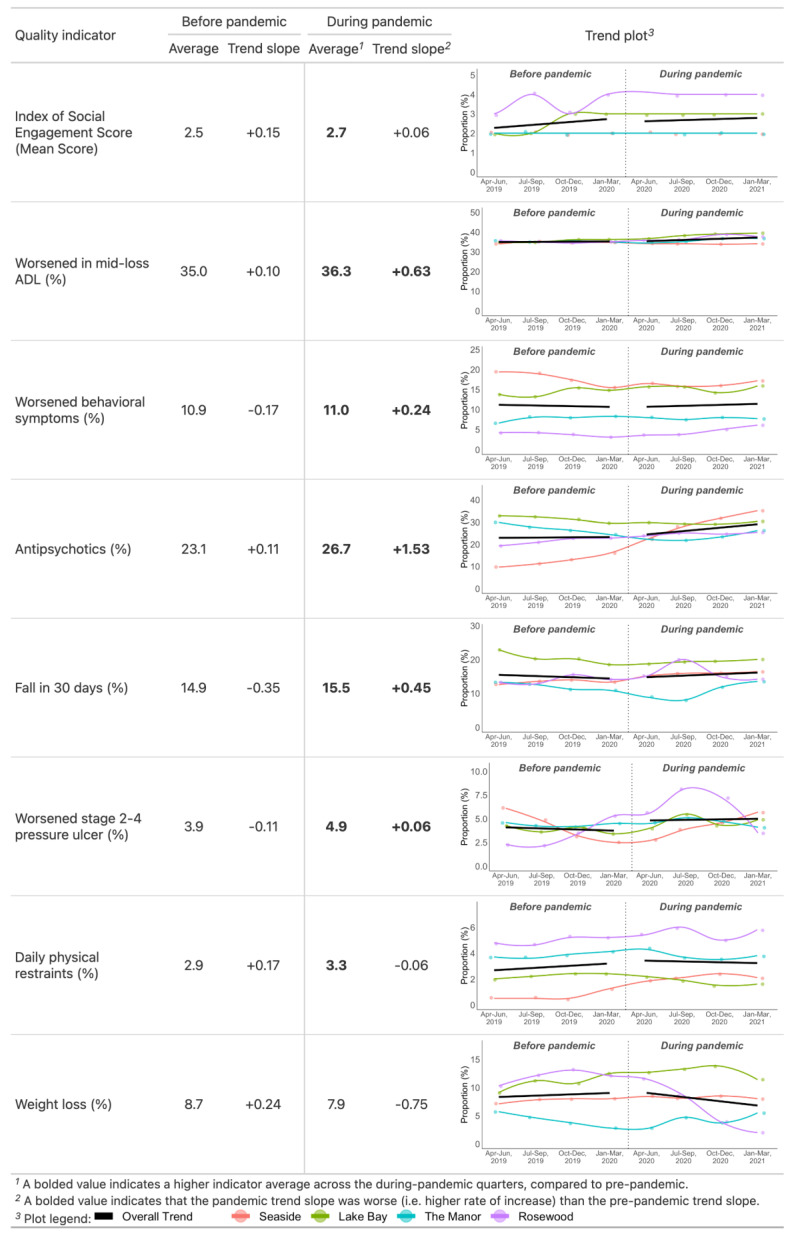
Overview of quality indicators with proportion averages, trend slopes, and trend plots for before-pandemic and during-pandemic quarters.

**Table 1 healthcare-11-03190-t001:** Characteristics of partner long-term care (LTC) homes and interview participants.

	LTC Homes				Details
	LTC #1	LTC #2	LTC #3	LTC #4	
Pseudonym	Seaside	Lake Bay	The Manor	Rosewood	
Municipality	Mission	Vancouver	Richmond	Vancouver	
# of staff	~250	~400	~280	~160	
# of residents	~151	~250	~250	~130	
Resident Interviews	(n = 2)	(n = 2)	(n = 2)	(n = 0)	Age 61–92 Female n = 2, Male n = 4 Lived in home for 1 to 20 years
Family Interviews	(n = 2)	(n = 4)	(n = 2)	(n = 2)	Age 56–74 Female n = 9, Male n = 1 Relationship to residents: Daughter, wife, husband
Staff Interviews	(n = 5)	(n = 5)	(n = 5)	(n = 3)	n = 14 > 40 years old Female n = 17, Male n = 1 Employed at home for 6 months to 37 years Job titles: Registered Nurse, Care Aide, Laundry Aide, Chef, Housekeeping Services
Leadership Interviews	(n = 2)	(n = 4)	(n = 2)	(n = 2)	Female n = 8, Male n = 2 Job titles: Chief Executive Officer, Executive Director, Nurse Manager, Care Aide Manager, Director of Human Resources, Clinical Operations Supervisor

**Table 2 healthcare-11-03190-t002:** Definitions of resident data indicators.

Resident Data Indicator	Definition
**Resident demographics**	
Sex (female)	The proportion of active residents that were female expressed as a percentage.
Age	Mean age of an LTC home’s active residents in years. Age is calculated based on residents’ age at the midpoint of each quarter.
Stay length	Mean stay length of an LTC home’s discharged residents in days or converted to years.
**Resident behaviours**	
Index of Social Engagement	Mean Index of Social Engagement (ISE) score for an LTC home’s assessed residents.
**Quality indicators ***	
Worsened/remained dependent in mid-loss ADL	Incidence indicator; the proportion of assessed residents that had worsened or remained dependent in mid-loss activities of daily living (ADLs) in percentage as representation of functional dependence.
Worsened behavioural symptoms	Incidence indicator; the proportion of assessed residents that had worsened behavioural symptoms in percentage.
Antipsychotics	Prevalence indicator; the proportion of assessed residents that had taken antipsychotics without a diagnosis of psychosis in percentage.
Fall in 30 days	Prevalence indicator; the proportion of assessed residents that had fallen in the last 30 days in percentage.
Worsened stage 2 to 4 pressure ulcer	Incidence indicator; the proportion of assessed residents that had stage 2 to 4 pressure ulcers that had worsened from the last assessment in percentage.
Daily physical restraints	Prevalence indicator; the proportion of assessed residents that were in physical restraints daily in percentage.
Weight loss	Prevalence indicator; the proportion of assessed residents that had had weight loss in percentage.

* Incidence indicators compare the latest assessments from the target quarter against the latest of the previous quarter and prevalence indicators describe residents based on their latest assessments in the quarter.

## Data Availability

Data are available upon reasonable request and approval from the Research Ethics Boards of the University of British Columbia.

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
