# Peer review of "Resident Impact of the Single Site Order Restricting Staff Mobility across Long-Term Care Homes in British Columbia, Canada"

_healthcare, 2023, doi:10.3390/healthcare11243190_

Round 1
Reviewer 1 Report
Comments and Suggestions for Authors
Overall, the manuscript is promising, and with some revisions and refinements, it can make a valuable contribution to the related fields. The following are some suggestions provided, I hope they can be helpful for your revisions.
l In the introduction, please describe the objectives of this study to help readers understand and anticipate the content of the study, especially the specific details regarding impact.
l In Table 1, the total number of staff interviewed in the 4 LTCs was 9, but in the details, it mentioned 'n=14 >40 years old, Female n=17, Male n=1,' which showed a discrepancy. The authors should review and correct this.
l Some quantitative indicators showed significant differences in performance among LTCs. If the authors have sufficient information, they should provide more detailed descriptions.
l The authors highlighted the interconnectedness between SSO, staffing challenges, and resident health deterioration. Strengthen this by explicitly discussing the cause-and-effect relationships between each factor. For instance, explicitly outline how the staffing challenges directly led to specific adverse outcomes for residents.
l While the authors mentioned the experiences of other countries like Denmark and Germany in dealing with staffing issues, consider drawing more explicit comparisons between their approaches and those in Canada. Discuss how Canada's responses to staffing challenges during the pandemic differed or were similar to these countries and the potential impacts of these differences.
l The authors might provide specific policy interventions or recommendations that could address staffing challenges effectively.
Reviewer 2 Report
Comments and Suggestions for Authors
First, I want to thank the journal for the opportunity to review this manuscript, which addresses such an important issue as the effect of healthcare policies to alleviate the course of the COVID pandemic and their impact on socio-healthcare attention from patients, family, and staff perspectives. I hope my comments will suggest improvements to an exciting research work while complementing the feedback from the rest of my fellow reviewers.
- The research team states that in-depth interviews were conducted regarding the qualitative part of this mixed-methods study. However, the design or theoretical perspective is not specified. I humbly suggest explicitly stating the type of qualitative research conducted (descriptive phenomenology, hermeneutics, etc.).
- Were any exclusion criteria considered besides the broad inclusion criteria implemented during participant recruitment?
- The terms in-depth and semi-structured interviews are used interchangeably in the manuscript, although they refer to different qualitative data collection techniques. As I understand it, both terms actually denote semi-structured interviews since an interview guide accompanied them. Please standardise the terminology based on whether they were actually in-depth interviews or semi-structured, and justify the choice appropriately.
- I suggest using a checklist that enhances transparency and quality in reporting qualitative studies, such as COREQ or SRQR.
- I suggest creating a section dedicated to rigour and quality, in which criteria such as those of Lincoln & Guba or Calderón (for instance) are argued, in addition to the series of strategies employed to ensure compliance with these criteria.
Round 2
Reviewer 2 Report
Comments and Suggestions for Authors
Congratulations on your modifications regarding parts sensitive to be improved. I wish you a successful career and research.